# Top-Down Detection of Eating Episodes by Analyzing Large Windows of Wrist Motion Using a Convolutional Neural Network

**DOI:** 10.3390/bioengineering9020070

**Published:** 2022-02-11

**Authors:** Surya Sharma, Adam Hoover

**Affiliations:** Department of Electrical and Computer Engineering, Clemson University, Clemson, SC 29634, USA; s@suryasharma.com

**Keywords:** automated dietary monitoring, eating detection, m-health, wearables, obesity

## Abstract

In this work, we describe a new method to detect periods of eating by tracking wrist motion during everyday life. Eating uses hand-to-mouth gestures for ingestion, each of which lasts a few seconds. Previous works have detected these gestures individually and then aggregated them to identify meals. The novelty of our approach is that we analyze a much longer window (0.5–15 min) using a convolutional neural network. Longer windows can contain other gestures related to eating, such as cutting or manipulating food, preparing foods for consumption, and resting between ingestion events. The context of these other gestures can improve the detection of periods of eating. We test our methods on the public Clemson all-day dataset, which consists of 354 recordings containing 1063 eating episodes. We found that accuracy at detecting eating increased by 15% in ≥4 min windows compared to ≤15 s windows. Using a 6 min window, we detected 89% of eating episodes, with 1.7 false positives for every true positive (FP/TP). These are the best results achieved to date on this dataset.

## 1. Introduction

Self-monitoring of eating episodes is important for weight loss and weight maintenance [1,2]. Traditionally, eating episodes are logged using manual methods such as food diaries (paper or electronic), which require human input [3]. However, methods based on manual input are prone to errors arising from human bias [4], and suffer from issues of long-term compliance [3]. A retrospective analysis of approx. 190,000 users of a food diary app found that less than 3% of the users continued using the app for more than one week [5]. For meal logging methods to be widely adopted, a critical barrier is to make them as easy to use as possible [3].

Automatic sensor-based methods can help self-monitoring by easing the user burden [6]. Previous works have tested a variety of sensor modalities positioned at different body locations [7]. Eating can be detected by analyzing breathing signals using an inductive plethysmography sensor positioned on the torso [8], throat movements using a piezoelectric sensor positioned at the neck [9], temporalis motion using an electromyogram sensor positioned at the temple [10], wrist motion using an inertial measurement unit (IMU) sensor positioned at the wrist [11], or a combination of different sensors across body positions [12,13]. Devices that have been tested include an earphone-like device [14], necklaces [15,16], a wearable device mounted behind the neck [17], eyeglasses mounted with sensors [10,18,19], and smartwatches [11,20,21,22,23]. Two surveys (N = 96, N = 55) showed that users preferred smartwatches over necklaces, earpieces, and headsets, and perceived them as more socially acceptable [24,25]. In this work, we describe a new approach based on tracking wrist motion that can be implemented in a smartwatch.

Eating is a complex task with high variability compared to some other human activities of daily life, such as walking or running. Figure 1a shows an example of 90 s of IMU data (accelerometer x, y, z, gyroscope yaw, pitch, roll) recorded by a wrist-mounted sensor while the subject was walking. Walking causes cyclical oscillations that can be easily recognized in a brief window of data. Figure 1b shows an example of 90 s of IMU data recorded by a wrist-mounted sensor while the subject was eating. Three bites (moments of ingestion) are labeled, along with two periods of food preparation (e.g., cutting and stirring). Eating tends to show large wrist rotation motions with smaller linear motions, which helps recognition [11]. However, general eating has more complexity and variability than walking, and thus requires more sophisticated methods for recognition.

Neural network (NN) classifiers are having a tremendous impact in the fields of computer vision, healthcare, and human activity recognition [26,27,28]. However, a recent survey of research in automated dietary monitoring found only five works that used deep learning [29]. Fontana et al. [30] extracted 68 features over 30 s windows from multiple sensors (jaw motion sensor, hand proximity sensor, chest accelerometer), and then processed this information using an NN [30]. Gao et al. extracted 18 features from sound data collected by off-the-shelf Bluetooth earphones to detect periods of eating [31]. Both these methods first extracted features using traditional techniques and then used an NN for classification. Kyritsis et al. developed an end-to-end deep learning approach using wrist motion data [22]. It uses two neural networks. The first network detects sub-components of hand-to-mouth gestures (raising hand, lowering hand, etc.), while the second network recognizes complete hand-to-mouth gestures (henceforth called bites). Experiments conducted in a laboratory setting found that the method could detect bites with an F_1_ score of 0.91 [22]. In more recent work, they tested an end-to-end network in free living conditions [23]. A convolutional neural network (CNN) was trained to look at 5 s windows of IMU data and predict the probability of the window containing a bite. Hypothesizing that the density of detected bites is high during meals, and low outside periods of eating, they identified periods of time during the day as eating with 95% weighted accuracy on their own dataset [23] and 79% on a larger dataset [32].

The bottom-up approach taken by previous works is demonstrated on 90 s of IMU data in Figure 2a. Three bites (moments of food intake) are labeled. A 5 s model is trained to look for individual bites and outputs the probability of a bite happening at each moment in time. Bite detections are indicated in the output by their high probabilities. To detect eating, bite detections are clustered over a period of time to identify a period of eating (a meal). Note that this model has no knowledge of other gestures that occur during eating, such as food preparation and manipulation (e.g., stirring, cutting, peeling fruit, etc.). Previous work has shown that modeling these gestures can help to improve the performance of detecting individual bites [33,34]. The 5 s model also has no knowledge of the periods of rest that normally occur after every ingested bite while food is masticated and swallowed. Modeling this information may also improve detection performance [35]. Finally, a bottom-up model is sensitive to the detection of other gestures that may resemble bites that can occur throughout the day during other activities. In contrast, our top-down approach to eating analyzes a longer period of time, learning all gestures and gesture sequences related to eating. Figure 2b illustrates the idea on the same 90 s of IMU data as in Figure 2a. Although the figure demonstrates the use of a 60 s model, we systematically test the effect of window size on recognition accuracy in our experiments. The novelty of this work is as follows:

We use a top-down approach to process large windows (0.5–15 min) of data (Figure 2b), in contrast to the bottom-up method prevalent in the literature, where short events such as a step or a bite are detected first (Figure 2a).We are the first group to demonstrate the performance of deep learning on a large dataset (4650 h of data in this work vs. 20 h of wrist motion data in previous works).Our classifier is evaluated against a dataset containing significantly more variability than that in previous works.

## 2. Methods

Figure 3 provides an overview of our method. We use end-to-end deep learning, where raw sensor data are used as the input instead of hand-crafted or extracted features. A window of *W* min is slid *S* s through a full day of 6-axis motion sensor data. This window is processed using a pre-trained CNN to determine the probability of eating p(t). We use a hysteresis algorithm to detect eating episodes of arbitrary length since the probability of eating is calculated in a fixed-size window. Figure 4 demonstrates the method. An episode is detected if the probability of eating is higher than threshold $T_{S}$ (start meal). A detected episode is ended if the probability of eating becomes lower than threshold $T_{E}$ (end meal). The use of two thresholds for hysteresis helps to smooth episode detections in a manner similar to button debouncing [36]. The following sections describe the dataset used in our work, our CNN design, the hysteresis algorithm, and the metrics used to evaluate our methods.

### 2.1. Clemson All-Day Dataset

The Clemson all-day (CAD) dataset contains 354 days of 6-axis motion sensor data from participants who wore a Shimmer3 motion tracker on their dominant wrist (e.g., right wrist for right-handed people) for an entire day [35]. Of 354 days, 351 were recorded by different subjects, so more than 99% of the data come from unique individuals. Participants self-reported the start and end times of meals using a button on the Shimmer3. The self-reports were reviewed with each participant the day after data collection. A total of 4680 h of wrist motion was recorded, containing 1063 eating episodes (meals or snacks). Each recording in the dataset contains 3-axis acceleration (accelerometer) and 3-axis angular velocity (gyroscope) data sampled at 15 Hz. Table 1 reports the summary statistics for the meals in the dataset. This dataset is available at http://www.cecas.clemson.edu/~ahoover/allday/ (accessed on 10 December 2021), and more details are available in [35].

### 2.2. Pre-Processing

Each recording in CAD is provided as a binary file containing floating point numbers in a flat sequence: $\{a_{1,x},\;a_{1,y},\;$$a_{1,z},\;\omega_{1,x},\;\omega_{1,y},\;\omega_{1,z},\;a_{2,x},\;a_{2,y},\;a_{2,z},\;\omega_{2,x},\omega_{2,y},\;\omega_{2,z},$$\ldots \;a_{n,x},\;a_{n,y},\;$$a_{n,z},\;\omega_{n,x},\;\omega_{n,y},\;\omega_{n,z}\}$, where $a_{n,x}$ represents the linear acceleration on the *x* axis at time *t* = *n* datum, and $\omega_{n,x}$ represents the rotational velocity on the *x* axis at time *t* = *n* datum. To reduce sampling noise, we smooth the data on each axis individually using a Gaussian filter. We use a value of σ=10 samples, which was used in previous work on the same dataset [35]. Finally, all axes are independently normalized using means and standard deviations using the z-norm in preparation for NN training.

### 2.3. Neural Network Classifier for Classifying Windows as Eating or Non-Eating

A CNN is used to provide a probability of eating or non-eating p(t) for a given window of 6-axis motion sensor data *W* min long. The design for this CNN is shown in Figure 5. The network has three 1D convolutional layers, followed by a global pooling layer, a dense fully connected layer, and a final output layer. All layers use a relu activation [37], except for the last layer, which uses a sigmoid activation [38] to output the class probability (eating vs. non-eating). All convolutional layers use a stride of 2 units and use the L1 norm for regularization [39]. We tested many variations in the number and types of layers, number of filters, and stride sizes. The final design was chosen based upon the accuracy achieved vs. model complexity. The model shown in Figure 5 contains approx. 7500 trainable weights.

The input to the CNN is *W* min (W×60 s) of 6-axis motion data. Our first convolutional layer uses $f_{1}=10$ 1D kernels, each of length $L_{1}=44$ units equivalent to 3 s of data at 15 Hz. The second convolutional layer uses $f_{2}=10$ 1D kernels of length $L_{2}=20$. The third layer uses $f_{3}=10$ 1D kernels of length $L_{3}=4$. The resulting layer is pooled using global average pooling to $f_{g}=10$ numbers, which feeds into a fully connected layer with $f_{c}=200$ units. The last layer contains 1 unit activated by the sigmoid function.

We used Keras [40] to develop and train our CNN models. We used binary cross-entropy loss [41] and the commonly used Adam optimizer [42] with a learning rate of 0.0001. A batch size of 32 was used during the training process. For each window size *W*, the model was trained for 150 epochs, after which the model weights were used for predictions on a separate test set. Training accuracy for all models reached a plateau within 150 epochs. Training was performed on Clemson’s Palmetto High-Performance Cluster, with each node using 125 GB of RAM and NVIDIA Tesla K40 GPUs.

### 2.4. Eating Episode Detector

For each day of data, and for each window size *W*, a continuous probability of eating p(e) is generated as the output of the neural network. An example is shown in Figure 3. This is done by sliding a window of size *W* minutes through the entire day of data at a slide of S=1 datum (1/15 s). Values for p(e) range between 0 (non-eating) and 1 (eating). The value at time index *t* corresponds to the center datum of the sliding window.

We identify periods of eating using a hysteresis-based detector to reduce the effect of noise in the p(e) signal [36]. The pseudo-code for the detector is shown in Algorithm 1. The inputs are p(e), the probability of eating at time *t* output by the NN, and $T_{S}$ and $T_{E}$, the hysteresis thresholds. The output is *s*, the count of detected episodes, and the arrays startIndex[s] and endIndex[s] denoting the detected boundaries. When p(e) rises higher than $T_{S}$ (start threshold), we mark the start of a detected eating episode. When p(e) drops to less than $T_{E}$ (end threshold), we mark the end of an eating episode. The use of two thresholds serves two purposes. First, it helps to reduce false positives by requiring a strong probability (>$T_{S}$) for at least part of the detection. Second, it helps to improve boundary precision by allowing for a weaker probability at the end of the detection. Eating can sometimes be more vigorous at the beginning of a meal, fading with satiety [35,43]. For example, a person may slow eating with fewer ingestion events towards the end of a meal. The second threshold $T_{E}$ allows for a lower probability towards the end of a detected episode. Finally, as all meals in CAD are longer than 1 min in length, we combine any two segments that are within 1 min of each other, and remove any segments shorter than 1 min to help avoid oversegmentation.
**Algorithm 1:** Pseudo-code for hysteresis-based detector used to segment eating episodes. 
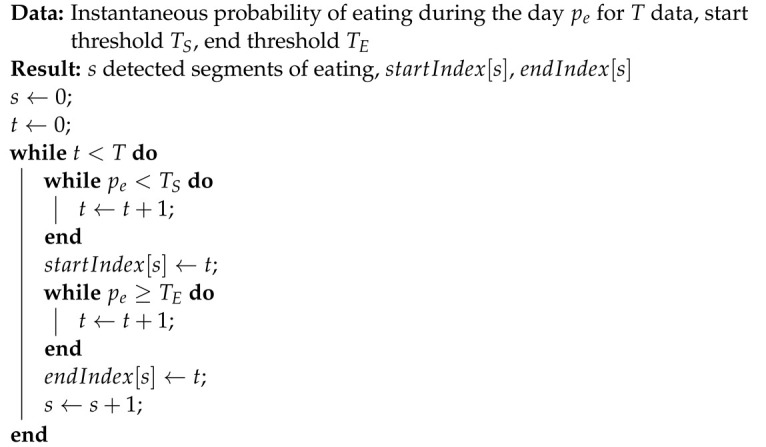



### 2.5. Evaluation Metrics

We report two types of metrics that can be used to evaluate the detection of eating episodes: episode metrics and boundary errors [44,45].

#### 2.5.1. Episode Metrics

Episode metrics describe success at detecting meals. The unit is a meal/snack. These measures quantify success at detecting daily patterns of eating (e.g., three meals at 8 AM, 12 PM, and 7 PM) while ignoring the total time spent eating. Figure 6 shows how we label detected segments as true positive (TP), false positive (FP), and miss (false negatives). A meal with any amount of overlap with a detected segment is counted as a TP. A meal with no overlap with a detected segment is counted as a miss. A detected segment not overlapping any meal is counted as an FP. True negatives (TN) cannot be defined [23]. While it is possible to calculate precision and recall, it is more intuitive to report TPR and FP/TP. These describe the percentage of meals successfully detected and the number of false alarms per true detection. In practice, TPR needs to be high and FP/TP needs to be low for a user to avoid the “cry wolf” effect [46,47].

#### 2.5.2. Boundary Metrics

Boundary metrics describe success at identifying when eating episodes started and ended [45]. The unit is minutes. They are only calculated for meals that are TPs (Figure 6). We report the average difference between the meal start time and detected segment start time as the start boundary error, and the average difference between the meal end time and detected segment end time as the end boundary error. In the special case of self-reported meals overlapping multiple detected events (Figure 6b,c), for each self-reported meal, we use the start/end of the first/last overlapping detection.

### 2.6. CNN Training

For training, windows of length *W* min with a slide of *S* = 15 s are generated for each day of data. These windows are labeled as eating (label = 1) if more than 50% of the window overlaps with a self-reported meal; otherwise, the window is labeled non-eating (label = 0). We found that a smaller slide *S* did not improve the classification performance, but significantly increased training times by creating more windows. Window generation yields a different number of windows depending on the hyperparameter *W*. For example, a window size of *W* = 6 min yields 57,907 windows of eating and 1,057,104 windows of non-eating. We balance the number of windows using undersampling of the non-eating class. We chose undersampling as oversampling did not show any improvement in performance. The full data contain approx. 5% eating and 95% non-eating, which is representative of everyday life. We use all the windows of eating and randomly select 5% of the windows of non-eating to balance. For example, at a window size of *W* = 6 min, this yields 57,907 windows of eating and 57,907 windows of non-eating for training.

### 2.7. CNN Validation

We use 5-fold cross-testing for validation of our model. Cross-fold validation reduces potential bias in the selection of the test set [48]. The model is trained on 80% of the data and tested on the remaining 20% of data. This is repeated five times, changing which 20% of the data is excluded during training and used for testing. The data were split by subject so that all data from the same subject were used either in training or testing. Figure 7 outlines the cross-validation process used, which consists of cross-folding, window cutting, and data balancing for model training and validation.

### 2.8. Hyperparameter Selection

The hysteresis-based segmenter uses thresholds $T_{S}$ and $T_{E}$ to identify periods of time considered eating activity. Combined, window size *W*, $T_{S}$, and $T_{E}$ can be considered hyperparameters. To search the hyperparameter space for *W*, $T_{S}$, and $T_{E}$, we use average cross-validation scores, as recommended in [49,50]. The advantage of a cross-validated search compared to a single train/validate/test search is that it generalizes better by evaluating multiple splits of the data [51]. The three hyperparameters have a complex interaction with the performance metrics TP, FP, and boundary error, and so we tune one hyperparameter at a time in relation to its most relevant performance metrics.

## 3. Results

In this section, the performance of the CNN during cross-validation is first used to identify general trends and select the hyperparameter value *W*. After this value is selected, the start threshold $T_{S}$ and end threshold $T_{E}$ for the hysteresis-based segmenter are discussed and selected. Finally, the performance of the overall method using the selected hyperparameters is reported.

### 3.1. Effect of Window Size W

A CNN is trained to process *W* minutes of data and generate the probability of eating p(e). Figure 8 shows the effect of varying *W* on the accuracy of classifying individual windows. Large window sizes (4–15 min) yield much higher classification accuracy (81–82%) than small window sizes (7.5–15 s, 66–69%). Previous work in human activity recognition has found a similar improvement as window size increases in identifying periods of time containing walking [28]. This result demonstrates that analyzing large windows of time to detect wrist motions associated with eating provides greater accuracy compared to analyzing a smaller window that seeks to detect individual hand-to-mouth gestures. We believe that this is because longer windows have additional important context on what happens between ingestion events. For evaluation of the detector at the eating episode level, we selected the value *W* = 6 min because larger window sizes did not produce a notable increase in NN accuracy. This is likely due to a tradeoff between analyzing a long enough period of time that it covers intergesture sequencing that occurs while eating [33], while at the same time not extending too long and thus capturing gestures that are outside the meal period (especially for short eating episodes such as snacks).

### 3.2. Effect of $T_{S}$ and $T_{E}$ on Evaluation Metrics

Once p(e) is generated for a full day of data, a hysteresis-based detector segments periods of time as eating. These segments can be labeled as true positives (TP) and false positives (FP), and the true positive rate (TPR) can be calculated. Figure 9 shows the effect of varying start threshold $T_{S}$ on TPR and FP/TP. *W* = 6 min and $T_{E}$ = 0.3 are kept constant. As expected, lower values of $T_{S}$ cause more detections. This leads to more TPs and FPs. Figure 10 shows the effect of varying $T_{E}$ on the average boundary error and FP/TP, while *W* = 6 min and $T_{S}$ = 0.8 are kept constant. We see a general trend where low $T_{E}$ values yield larger boundary errors but fewer FPs because individual detections are longer. The values $T_{S}$ = 0.8 and $T_{E}$ = 0.4 are selected as they balance the TPR, FP/TP, and boundary error metrics.

### 3.3. Eating Activity Detection Performance

Of 1063 total meals in the CAD dataset, we detected 944 meals, missed 119, and triggered 1650 FPs. This results in a TPR of 0.89 with 1.7 FP/TP (or ≈4.7 FPs/day). Table 2 compares the performance of our method against previous works, showing that our new method achieved a comparable TPR with a significant reduction in the number of false positives. We believe that this is because the analysis of longer windows of time helps to classify sequences of wrist motion that may briefly resemble eating (such as a small amount of hand-to-head gesturing), but that with longer surrounding context can be correctly classified as non-eating.

The start boundary error of our method is −1.5 ± 4.5 min, meaning that, on average, detected segments start 1.5 min before the self-reported start time for a meal. This is a 66% reduction in error compared to previous work on CAD where the start boundary error was −4.5 ± 14 min [35]. The end boundary error is 0.9 ± 4.9 min, meaning that the end of detected meals came, on average, 0.9 min after the end of the corresponding self-reported meals. This is a 88% reduction in error compared to previous work on CAD where the end boundary error was 7.3 ± 11 min [35]. Thus, the analysis of longer windows of time did not adversely affect the precision in detecting the boundaries of eating episodes, and in fact it improved their detection.

### 3.4. Comparative Results

In Table 2, it can be seen that most previous works analyzed very short windows of time (5–6 s) looking for individual hand-to-mouth gestures, and then aggregated them to identify periods of eating [23,32,52]. Only two methods previously explored using longer windows of time by segmenting data into windows of varying length [11,35], but neither used a neural network for analysis and they detected far more false positives than our new method.

Table 2 shows that many previous works do not report metrics for accuracy in detecting eating episodes, opting instead to focus on evaluating accuracy at classifying each individual datum as eating or not eating. We therefore include Table 3 to allow for further evaluation of our work. Compared to previous work on the large CAD dataset [35], our new method achieved higher weighted accuracy and a much higher F1 score. Some of the studies shown in Table 3 reported better F1 scores at this fine-grained level of accuracy, but it is important to emphasize that these experiments were conducted on limited data. For example, the bottom-up classifier in [23] was trained on laboratory data for 12 people eating a single meal each, in which eating was restricted to using a fork, knife, and spoon, and no liquid intake was allowed. It was then tested on data collected by six people outside the lab having the same restricted eating conditions, achieving precision of 88% and recall of 92% (row 8 of Table 3). However, when the same method was tested on a different dataset collected by 11 people outside the lab, with less restricted eating conditions, the precision decreased to 46% and recall decreased to 63% (row 5 of Table 3). This demonstrates why it is important to test on a dataset collected by a much larger number of subjects, with unrestricted eating conditions. A dataset with a small number of subjects contains limited variability in daily activities, and a dataset with restricted eating contains limited variability in wrist motion patterns during eating. Additionally, eating in everyday life often occurs with secondary activities such as watching media, working, and talking with friends, which further increases the complexity of detecting eating [35]. Experiments with small datasets and/or restricted conditions should be interpreted with caution as they are not likely to yield similar results when repeated on larger, everyday-life datasets [45].

## 4. Discussion

The contributions of our work can be summarized as follows:We show a new method that can detect periods of eating by tracking wrist motion data. We take a top-down approach, analyzing large windows (0.5–15 min) to determine if eating occurred.We show evidence that considering longer periods of data to detect eating can improve performance by up to 15%.We train and test against the largest available dataset of its kind (10×–100× larger than other datasets), giving us confidence in our results.

Our top-down approach can be contrasted against a bottom-up approach that analyzes a small window (5 s) to find individual ingestion events, and then grouping them to identify the entire eating episode. The bottom-up approach is common in previous work, where activities such as running, walking, jumping, or cycling are classified by detecting individual steps or motions. Cyclical patterns in these activities are easy to model and can be detected easily. In contrast, cyclical patterns are less common during a meal, and wrist motion during a meal can exhibit a wide range of motion in addition to cyclical patterns. Our top-down approach can take advantage by modeling motions that occur *between* ingestion events, such as manipulating food in preparation for ingestion and resting between ingestion events [33]. Our results found a 15% improvement in identifying if larger windows contain eating compared to smaller windows. Episode detection using larger windows yielded fewer false positives than previous works, and improved the boundary detection of the start and end times of the episodes.

There is growing evidence that a very large dataset may be necessary to train a classifier that generalizes to a larger population. Several previous works have found large performance decreases when trained on small laboratory datasets and subsequently tested in free living conditions conditions [15,32,52,53]. A dataset containing few meals and few subjects has low within-subject and between-subject variability. This results in less variation in wrist motion patterns. Although the experiments reported in this work were tested on 10×–100× more data than most previous works, we believe that this may still not be enough data. When looking at our five-fold results, we see differences of up to 7% in window classification accuracy, 0.95 FP/TP, and 1.8 min in average boundary error across the five folds. These performance differences suggest that even more data may be needed and that k-fold cross-validation results need to be reported to evaluate generalizability.

Researchers are increasingly noticing the effect of secondary activities during eating, such as simultaneously walking while eating, watching media, and doing office work or chores [15,32,35,52,53]. Secondary activities can obscure eating in wrist motion patterns [35]. A study of the CAD dataset found that, on average, 12.8% of a self-reported eating episode consisted of resting and 5.5% consisted of walking [35]. This indicates that people can spend a sizable portion of what they consider a “meal” doing things other than eating. We believe that this provides further motivation to transition from bottom-up approaches to top-down approaches for eating episode detection. Data acquired with ground truth at a fine detail (e.g., with individual ingestion events marked) require more complex instrumentation and control of experimental conditions. This can adversely affect the naturalness of behaviors, causing subjects to avoid secondary activities while under observation. This leads to classifier training that does not generalize when tested on data collected in everyday life.

A limitation of our method is that it may not work well on brief periods of eating, such as snacking. Figure 11a shows a histogram of the lengths of meals in CAD. Of 57 meals of length less than 3 min, 24 (42%) are missed by our method. A trend of missing more shorter meals than larger meals can also be seen. Figure 11b shows the same histogram for our method using hyperparameters *W* = 2 min, $T_{S}$ = 0.75, and $T_{E}$ = 0.55. This reduces the number of missed snacks to just 10 (18%) and increases the episode detection rate to 95%. However, it also increases episode FP/TP to 6.1, which may not be acceptable in practical use. Mirtchouk et al. reported similar issues when detecting snacks, noting that 67% of their false positives were less than 3 min in length [32]. They concluded that a different model might be needed to detect snacks. We agree with their conclusion.

A second limitation of our top-down approach is that it does not identify individual consumption events that may be identified by a bottom-up approach. These need to be identified by additional analysis of the detected episodes. This is a topic for future work. A third limitation is that while our true positive rate is high (detecting nearly 90% of eating episodes), it still yields more than one false positive per true positive. In practice, this could be addressed by asking the user in real time if they were in fact eating or not after each detection. However, the frequency of these prompts would need to be low enough to avoid over-burdening the user. Future work needs to determine the amount of prompts that would be acceptable to users. Lastly, the CAD dataset only contains the start and end boundaries of self-reported meals, limiting the detection of activities to self-reported meals. Future data collections could record additional information such as the boundaries of meal preparation and cleanup to increase the number of activities that can be detected.

## 5. Conclusions

Finally, we conclude that further top-down approaches for meal detection should be explored, in contrast to the bottom-up approach prevalent in the literature [13,21,22,23,28,30,54]. Non-eating activities in proximity to eating, such as meal preparation and cleanup, may contain additional information valuable for the task of detecting eating episodes. The same approach may be applied to other research; for example, a top-down approach may be applied to detect activities such as watching TV, attending a class, visiting a gym, or using a toilet when monitoring diabetes patients for sedentary behavior or the frequency of using the toilet. Another use of the top-down approach could be in geographic information systems (GIS), where image data could be scanned to look for potential geographic regions of interest. For example, water bodies in the desert may be typically be surrounded by vegetation, and looking for both could improve the performance of finding either. Modeling and classification of these activities using neural networks with complex architectures may further improve the detection of eating episodes. We would also like to reiterate our belief that the field must continue to transition from small datasets to large datasets [45].

## Figures and Tables

**Figure 1 bioengineering-09-00070-f001:**
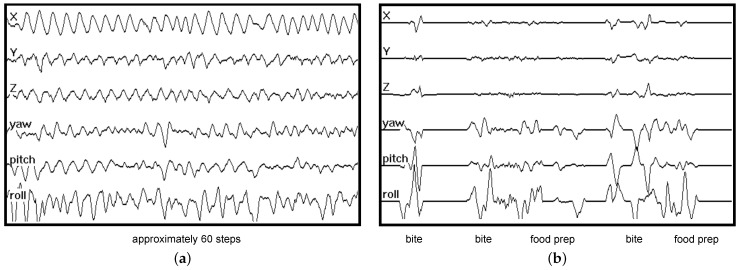
IMU data (accelerometer x, y, z, gyroscope yaw, pitch, roll) recorded by a wrist-mounted sensor. Walking causes repetitive oscillations and can be easily recognized in a brief window of data. Eating causes large rotational motion with limited linear motion but is not as easily recognized as walking. (**a**) 90 s of walking; (**b**) 90 s of eating.

**Figure 2 bioengineering-09-00070-f002:**
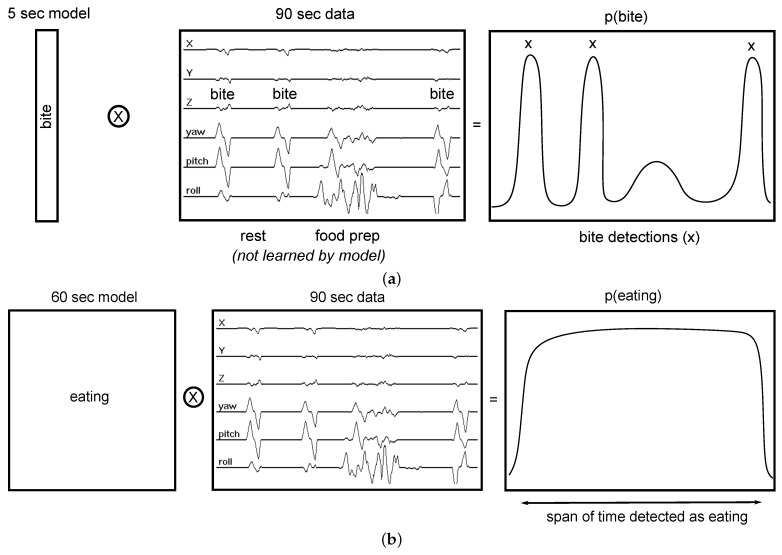
Comparing a bottom-up vs. top-down model for detecting eating. ⭙ implies convolution. A bottom-up model has no knowledge of gestures related to food preparation (e.g., stirring, cutting) and rest that take place between bites. A top-down model learns all these gestures and gesture sequences to improve its detection of when eating is occurring throughout the day. (**a**) A bottom-up model learns the appearance of a bite (duration 5 sec) and outputs the probability of its occurrence across the data. Individual bite detections are clustered to identify periods of eating (a meal). This model is sensitive to any gestures resembling bites, and has no knowledge of gestures that happen between bites (e.g., food preparation and rest during ingestion). (**b**) A top-down model learns all gestures and gesture sequences related to eating (duration 1 min shown). It outputs the probability of eating across the data.

**Figure 3 bioengineering-09-00070-f003:**
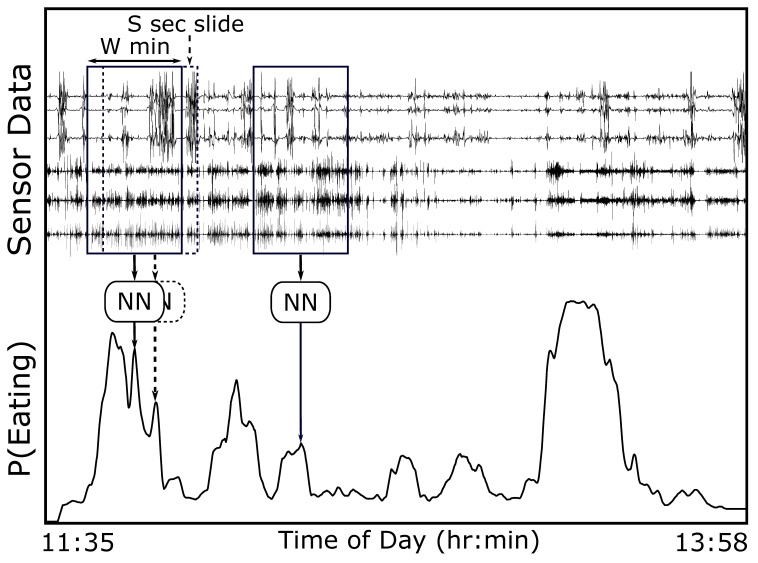
We use a sliding window of size *W* minutes and slide S s to generate a continuous probability of eating for the entire day (solid line).

**Figure 4 bioengineering-09-00070-f004:**
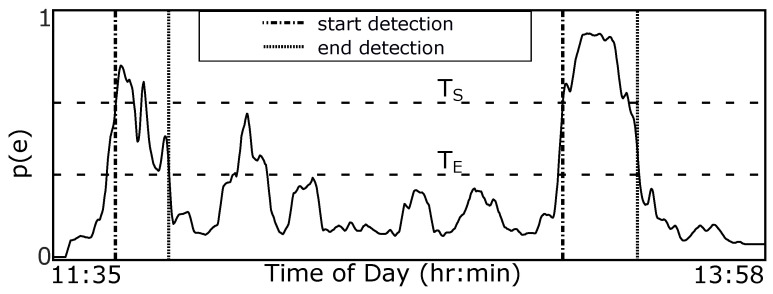
Probability of eating throughout the day is converted to periods of eating (vertical dashed lines) using a hysteresis-based detector with thresholds $T_{S}$ and $T_{E}$.

**Figure 5 bioengineering-09-00070-f005:**
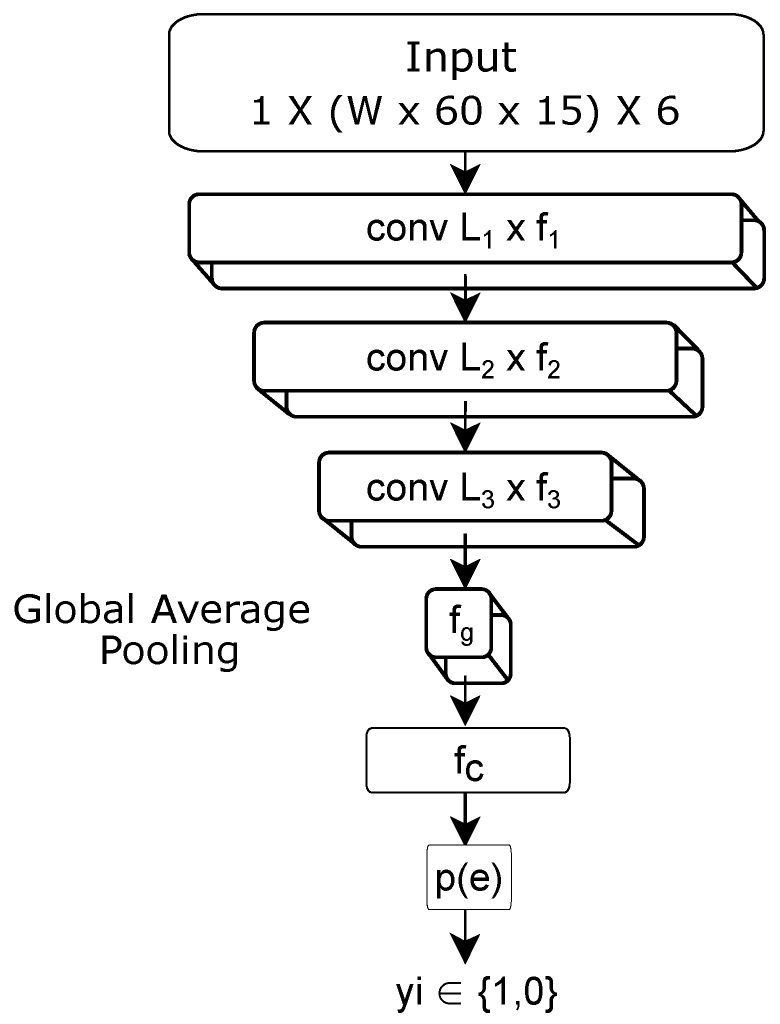
Neural network design.

**Figure 6 bioengineering-09-00070-f006:**
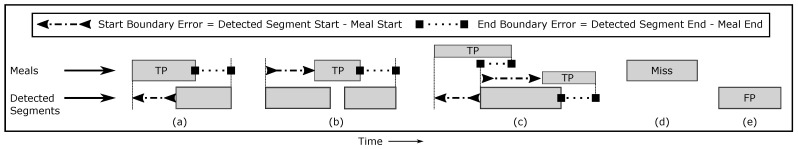
Evaluation at episode level: (**a**) one meal overlaps one detected segment, (**b**) one meal overlaps multiple detected segments, (**c**) multiple meals overlap one detected segment, and each meal is evaluated for its own TP and boundary error. (**d**) no detected segment overlaps a meal (miss), and (**e**) a detected segment does not overlap any meal (FP).

**Figure 7 bioengineering-09-00070-f007:**
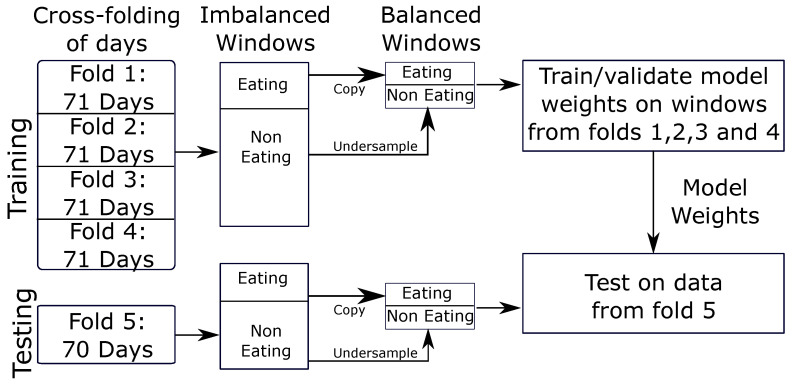
The above cross-folding process is repeated 5 times. Each fold is used as the test set once, while the other 4 folds are used for training.

**Figure 8 bioengineering-09-00070-f008:**
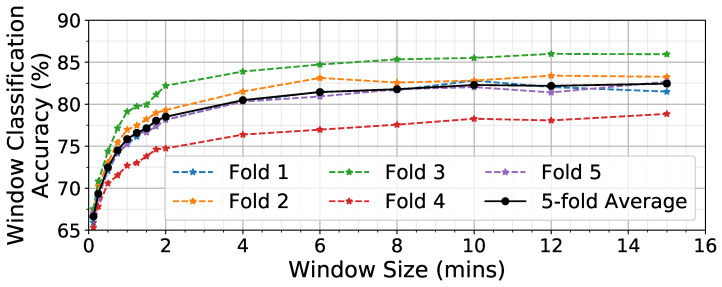
Effect of window size *W* on window classification accuracy.

**Figure 9 bioengineering-09-00070-f009:**
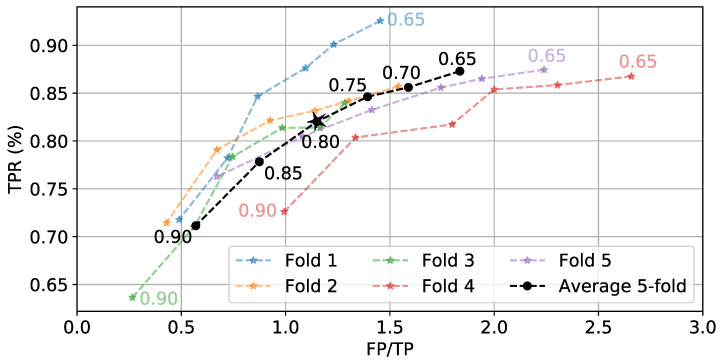
Effect of $T_{S}$ (numbers next to lines) on TPR and FP/TP while keeping *W* = 6 min and $T_{E}$ = 0.3 fixed. ⋆ indicates selected value.

**Figure 10 bioengineering-09-00070-f010:**
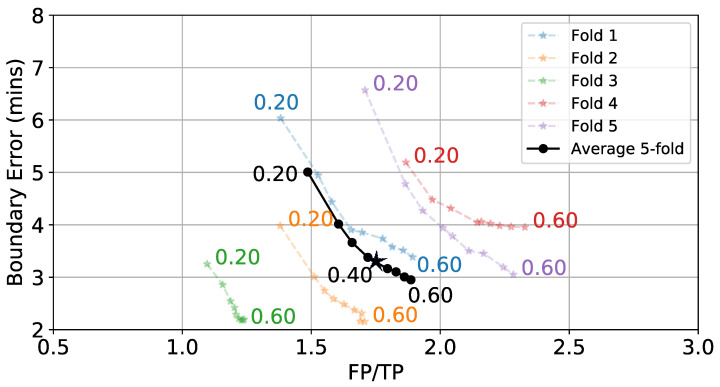
Effect of $T_{E}$ (numbers next to lines) on boundary error and FP/TP while keeping *W* = 6 min and $T_{S}$ = 0.8 fixed. ⋆ indicates selected value.

**Figure 11 bioengineering-09-00070-f011:**
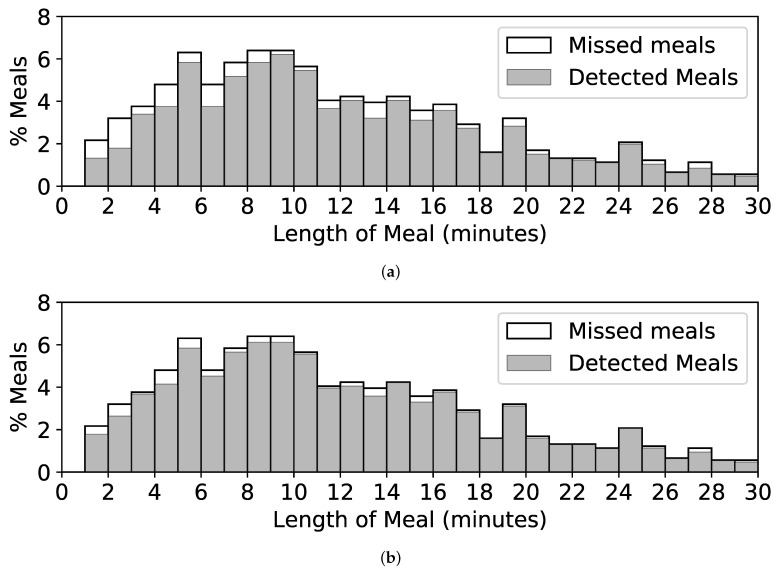
Histograms showing lengths of self-reported meals in CAD and the percentage of meals detected. (**a**) Detected and missed meals when using *W* = 6 min, $T_{S}$ = 0.8, and $T_{E}$ = 0.4. (**b**) Detected and missed meals when using *W* = 2 min, $T_{S}$ = 0.75, and $T_{E}$ = 0.55.

**Table 1 bioengineering-09-00070-t001:** Duration of self-reported eating events in the CAD dataset.

Meals	Total	Mean	Std	Median
1063	250 h	14 min	11 min	11 min

**Table 2 bioengineering-09-00070-t002:** Results for this work compared to related works (per detected meal). Most related works were tested on small datasets and meal detection metrics were not reported. Our new method achieves the best performance on the large CAD dataset.

Work	Dataset	Subjects	Meals	Approach	Window Size	TPR (%)	FP/TP
This Work	CAD [35]	351	1063	Top-down	6 min	89	1.7
Sharma, 2020 [35]	CAD [35]	351	1063	Top-down	varying	89	5.2
Dong, 2014 [11]	iPhone [11]	43	116	Top-down	varying	86	3.8
Kyritsis, 2020 [23]	ACE-E+FL [32]	11	86	Bottom-up	5 s	-	-
Mirtchouk, 2017 [32]	ACE-E [32]	6	55	Bottom-up	5 s	87	-
Mirtchouk, 2017 [32]	ACE-E/FL [32]	5	31	Bottom-up	5 s	94	-
Kyritsis 2020 [23]	FreeFIC (FF) [23]	6	17	5 s	-	-	
Kyritsis, 2020 [23]	FF held-out (FFHO) [23]	6	6	Bottom-up	5 s	-	-
Thomaz, 2015 [52]	Wild-7 [52]	7	-	Bottom-up	6 s	-	-
Thomaz, 2015 [52]	Wild-Long [52]	1	-	Bottom-up	6 s	-	-

**Table 3 bioengineering-09-00070-t003:** Results for this work compared to related works (per datum). Note that most related works were tested on much smaller datasets.

Work	Dataset	Total Hours	Eating Hours	Precision (%)	Recall (%)	TNR (%)	F_1_ Score (%)	*ACC_W_* (%)
ours	CAD [35]	4680	250	36	69	93	**48**	**80**
[35]	CAD [35]	4680	250	14	76	73	**23**	**77**
[11]	iPhone [11]	449	22	-	81	82	-	81
[23]	ACE-E+FL [32]	250	20	46	63	63	53	79
[32]	ACE-E [32]	144	12	25	83	-	38	79
[32]	ACE-E/FL [32]	254	20	31	87	-	-	85
[23]	FreeFIC [23]	77	5	88	92	99	90	96
[23]	FFHO [23]	35	2	86	94	99	90	96
[52]	Wild-7 [52]	32	2	67	89	-	76	-
[52]	Wild-Long [52]	422	16	65	79	-	71	-

## Data Availability

Data used for this study can be obtained at http://www.cecas.clemson.edu/~ahoover/allday/ (accessed on 10 December 2021), and more details are available in [35].

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
