# Peer review of "Top-Down Detection of Eating Episodes by Analyzing Large Windows of Wrist Motion Using a Convolutional Neural Network"

_bioengineering, 2022, doi:10.3390/bioengineering9020070_

Round 1

Reviewer 1 Report

The manuscript proposes the convolutional neural network for detecting  eating episodes. The proposed concept is new and relevant. However, major revisions, improvement and reading are essential before recommendation for publication.

One of the major revision suggestion is to expand the state of the art. This would support to better understand and research gap and contribution of the paper.

The other major issue, is that manuscript has no conclusions. Please add it.

The other major issue is the lack information on the validation. What validation had been used? Why CNN had been selected and what is its performance compared to other DL or ML methods?

Further minor comments include:

Although the paper has appropriate length and informative content, several parts must be improved and written in better grammar and syntax. It would be essential if authors would consider revising the organization and composition of the manuscript, in terms of the definition/justification of the objectives, description of the method, the accomplishment of the objective, and results. The paper is generally difficult to follow. Paragraphs and sentences are not well connected. Furthermore, I advise considering using standard keywords to better present the research.

Please revise the abstract according to the journal guideline. It must be under 200 words. The research question, method, and the results must be briefly communicated. The abstract must be more informative. I suggest having four paragraphs in the introduction for; describing the concept, research gap, contribution, and the organization of the paper. The motivation has the potential to be more elaborated. You may add materials on why doing this research is essential, and what this article would add to the current knowledge, etc. The originality of the paper is not discussed well. The research question must be clearly given in the introduction, in addition to some words on the testable hypothesis. Please elaborate on the importance of this work. Please discuss if the paper suitable for broad international interest and applications or better suited for the local application? Elaborate and discuss this in the introduction.

State of the art needs improvement. A detailed description of the cited references is essential. Several recently published papers are not included in the review section. In fact, the acknowledgment of the past works related to machine learning and deep learning by others, in the reference list, e.g., Modeling pan evaporation using Gaussian Process Regression K-Nearest Neighbors Random Forest and support vector machines; comparative analysis, can be studied. Consequently, the contribution of the paper is not clear. Furthermore, consider elaborating on the suitability of the paper and relevance to the journal. Kindly note that references cited must be up to date.    

Elaborate on the method used and why used this method.

Limitations and validation are not discussed adequately. The research question and hypothesis must be answered and discussed clearly in the discussion and conclusions. Please communicate the future research. The lessons learned must be further elaborated in the conclusion by discussing the results to the community and the future impacts. What is your perspective on future research?   

state of the art can be further improved to show the importance of CNN in general applications. Further reading on review of the CNN application is essential, e.g., Deep learning for detecting building defects using convolutional neural networks. CNN needs further reference e.g., List of deep learning models. 

Reviewer 2 Report

(1)In Section 1.1, the contributions should be summarized point-by-point, and then this section can be easily followed.
(2)CNN is suitable for a fixed size signal, such as image. Why not use RNN for the sequence signal?
(3)In Table 3, since the datasets are different for the different methods, how to ensure that the comparison is fair?
(4)The review of the related works and comparison experiments can be more sufficient. Please carefully read, cite and compare (if applicable) the following papers that are on light-weight deep learning models for medical classification and helpful to the revision.
- A light-weight practical framework for feces detection and trait recognition
- StoolNet for color classification of stool medical images
(5)In Section 3, the discussions should be summarized point-by-point, and then this section can be easily followed.
(6)This paper has application values, but the academic values is not high.

Reviewer 3 Report

This paper proposes the use of a DNN for eating detection using CNN. The paper is clearly explained and presented. The results are a bit low but comparable to state of the art. I suggest some comments to improve the paper.

  • Section 1.1 title, perhaps is not necessary. When there is only one subsection, I think it is not necessary to define this subsection title.
  • I’d like to ask how the ground truth was generated. Is it a 0/1 label? When do you decide that a given windows is an eating episode? When more than 50% overlap with an eating episode??
  • You used a K-fold cross-validation. Did you use a subject-wise strategy? I mean all the recording from the same subject were included in the same fold? This way we do not have recordings from the same subject in training and testing?
  • In tables 2 and 3 I’d suggest including similar metrics to better analyze and compare.
  • I think the results are not very good (in line 242 the FP is higher than 1600???) and the F-score is lower than 50%. I am wondering if there is a problem with the balance between the number of examples and the CNN complexity. Perhaps, could be interesting to consider smaller windows to have more data to train the CNN and then, combine the output scores of consecutive windows to provide a decision in longer periods of time. An interesting analysis has been performed in this paper: https://link.springer.com/article/10.1007/s11063-021-10611-w

Round 2

Reviewer 1 Report

Without a correct validation and providing a solid comparative analysis the manuscript cannot be accepted.

Author Response

Dear Reviewer,

We have expanded the description of our model validation process and include a new figure to show how we used 5-fold cross validation in section 2.7. We have also highlighted comparisons against other work in the newly added section 3.4.

Thank you

Reviewer 2 Report

Can be accepted.

Author Response

Thank you

Reviewer 3 Report

The authors have improved the paper significantly and it can be accepted

Author Response

Thank you, we appreciate your time.